# A New Interactive Tool to Visualize and Analyze COVID-19 Data: The PERISCOPE Atlas

**DOI:** 10.3390/ijerph19159136

**Published:** 2022-07-26

**Authors:** Daniele Pala, Enea Parimbelli, Cristiana Larizza, Cindy Cheng, Manuel Ottaviano, Andrea Pogliaghi, Goran Đukić, Aleksandar Jovanović, Ognjen Milićević, Vladimir Urošević, Paola Cerchiello, Paolo Giudici, Riccardo Bellazzi

**Affiliations:** 1Department of Electrical, Computer and Biomedical Engineering, University of Pavia, 27100 Pavia, Italy; enea.parimbelli@unipv.it (E.P.); cristiana.larizza@unipv.it (C.L.); riccardo.bellazzi@unipv.it (R.B.); 2Department of Biostatistics, Epidemiology and Informatics, Perelman School of Medicine, University of Pennsylvania, Philadelphia, PA 19104, USA; 3Telfer School of Management, University of Ottawa, Ottawa, ON K1N 6N5, Canada; 4Hochschule für Politik, The TUM School of Governance, Technical University of Munich (TUM), 80333 Munich, Germany; cindy.cheng@hfp.tum.de; 5Departamento de Tecnología Fotónica y Bioingeniería, Universidad Politècnica de Madrid, 28040 Madrid, Spain; manuel.ottaviano@upm.es; 6GeneGIS GI Srl, 20148 Milan, Italy; a.pogliaghi@genegis.net; 7Research & Development Department Belit Ltd., Trg Nikole Pašića 9, 11000 Belgrade, Serbia; goran.djukic@belit.co.rs (G.Đ.); aleksandar.jovanovic@belit.co.rs (A.J.); ognjen.milicevic@belit.co.rs (O.M.); vladimir.urosevic@belit.co.rs (V.U.); 8Dipartimento di Scienze Economiche e Aziendali, University of Pavia, 27100 Pavia, Italy; paola.cerchiello@unipv.it (P.C.); paolo.giudici@unipv.it (P.G.)

**Keywords:** COVID-19, pandemic, Atlas, public health, policy making, data mining, PERISCOPE

## Abstract

Since the start of the 21st century, the world has not confronted a more serious threat to global public health than the COVID-19 pandemic. While governments initially took radical actions in response to the pandemic to avoid catastrophic collapse of their health care systems, government policies have also had numerous knock-on socioeconomic, political, behavioral and economic effects. Researchers, thus, have a unique opportunity to forward our collective understanding of the modern world and to respond to the emergency situation in a way that optimizes resources and maximizes results. The PERISCOPE project, funded by the European Commission, brings together a large number of research institutions to collect data and carry out research to understand all the impacts of the pandemic, and create predictive models that can be used to optimize intervention strategies and better face possible future health emergencies. One of the main tangible outcomes of this project is the PERISCOPE Atlas: an interactive tool that allows to visualize and analyze COVID-19-related health, economic and sociopolitical data, featuring a WebGIS and several dashboards. This paper describes the first release of the Atlas, listing the data sources used, the main functionalities and the future development.

## 1. Introduction

The COVID-19 pandemic has been an unexpected event that took the world by surprise at the end of the year 2019. COVID-19 is a disease caused by the SARS-Cov-2 virus, a novel coronavirus that was first observed in China at the end of 2019 and quickly spread to the rest of the world in 2020. Most of the population lacked immune defenses against this new virus, and as such, both the hospitalization and death rates were significantly higher compared to those of other coronaviruses. Meanwhile, the virus’ rapid spread quickly led to a global health emergency, forcing most governments to take severe measures to avoid the overwhelming of hospitals. As of April 2022, COVID-19 has killed about 6.2 million people around the world, but this number is likely underestimated.

The sudden outbreak of this dangerous disease had a significant impact on the entire world’s population, changing not only the global health panorama, but also the socio-economic one. Indeed, governments took a myriad of measures to severely contain the virus, including restricting people from leaving their homes or gathering en masse, many of which disrupted both social and economic life. While research on these effects is still ongoing, several studies have shown for instance that social disparities in several areas of the world have been exacerbated [1,2], and also that mental health worsened for a large part of the population [3,4]. In some cases, these changes were associated with an increase in crime rate [5], and more generally to a change of life habits and human relationships that will likely last for years.

From a scientific point of view, the advent of the virus represented a unique opportunity for scientific research, particularly in the fields of health (e.g., for the creation of vaccines or to understand the contagion mechanisms) and socio-economics. In this context, PERISCOPE (Pan-European Response to the ImpactS of COVID-19 and future Pandemics and Epidemics) [6] has been leading the effort to measure and research the socio-economic and behavioral impacts of the pandemic since November 2020. PERISCOPE is a consortium of 32 European institutions funded by the EU Commission and is coordinated by the University of Pavia, Italy [7].

Following the necessity of collecting, storing and visualizing large quantities of COVID data to provide useful resources to monitor and contain the pandemic, numerous data visualization and analysis tools have been developed by researchers and private companies all around the world, making it possible to share large quantities of data and leading to a boost in academic research on the topic. On the other hand, the high number of different resources makes COVID information fragmented and difficult to navigate. Most tools and scientific studies focus on one specific aspect of the pandemic, e.g., prediction of epidemiological trends, characterization of economic impacts or social implications of the pandemic.

One of PERISCOPE’s main aims is to overcome this problem creating the European COVID-19 Atlas, i.e., a large collection of data, information and interactive tools that can be used to understand the impact of the virus in Europe by allowing users to visualize spatially-enabled data and to access advanced analysis tools. The PERISCOPE Atlas seeks to be a collection of heterogeneous datasets and analysis tools on several aspects of the pandemic, that can be used as the main point of reference by researchers and policy makers studying the progression of not only this pandemic but to be a resource for preparing for future similar emergency situations.

The main aim of this paper is to describe the first release of PERISCOPE’s COVID Atlas. As such, in what follows, we describe the PERISCOPE project in greater detail before then presenting the COVID Atlas in greater depth. To that end, we further describe the data sources that have been identified and used to create the repository, as well as the main functionalities of the Atlas itself, which is already available and open-access.

## 2. The PERISCOPE Project

The PERISCOPE project [6,7] has been funded by the European Commission in 2020 to undertake research on the new COVID-19 disease. An important core of this work is to create a comprehensive repository of data, tools, models, and interactive systems that can be used to visualize and create statistics about the COVID-19 pandemic, and to provide data about the socio-economic impact of the virus that can be used by policy makers to better tune future containment strategies. The PERISCOPE consortium is composed of 32 partners [8], including academic and other research institutions located throughout Europe, and coordinated by the University of Pavia, Italy. The partners of the consortium carry out theoretical and experimental research to contribute to a deeper understanding of the short- and long-term impacts of the pandemic and the measures adopted to contain it. Such research-intensive activities will allow the consortium to propose measures to prepare Europe for future pandemics and epidemics.

The main goals of the project can be summed up in the following points:Create the COVID-19 Data Atlas, in order to provide data, geographical and analysis tools that can be easily accessed and used as a point of reference for future research and public health decision making.Perform innovative statistical analysis and apply machine learning methods on the large amount of existing COVID data, to discover relations, patterns and create useful predictions regarding the disease.Identify successful policy practices adopted at a local level, which can be extended to the European or even global level.Develop guidance for policy makers to enhance Europe’s preparedness to possible future pandemics or similar events.

The project is organized in a vast number of work packages that address every aspect of the organization and management of data gathering, integration and analysis. A number of dimensions are covered, including impacts on health systems, socio-economic impacts, mental health and inequalities, health policies, behavioral science, governance and education.

The Data Atlas can be considered one of the core components of the PERISCOPE infrastructure, as it allows to navigate the large amount of collected data, inspect public health indicators related to the pandemic and perform statistical analyses.

## 3. Data Sources

Numerous data sources have been used, and other will be integrated, to create the Data Atlas and allow various visualizations and analyses. The data collected can be categorized in mainly three groups:Health Data, i.e., measures that concern the progression of the pandemic in terms of number of cases, hospitalizations, hospital occupancy, positive-tested ratio, recoveries, incidence and deaths. The aim of these variables is to measure the impact of the pandemic on the general population’s health and the local healthcare systems.Economic Data, i.e., monitoring of common economic indicators such as GDPs, employment rate, import–export rate, etc. The aim of this data is to show the impact of the pandemic, and of the containment measures and behavioral changes related to it, on the world’s economy.Social and Political Data, i.e., data related to policies and regulations issued by the local governments to contain the spread of the virus, together with population behavioral changes directly or indirectly related to them.

The rest of this section lists the main data sources used in the Atlas so far, and briefly shows other sources that were identified for future integration.

### 3.1. European Center for Disease Control

The European Centre for Disease Control and Prevention (ECDC) is an agency that is part of the European Union, created to improve Europe’s defenses against infectious diseases [9]. It was established in 2004 with headquarters in Sweden and its tasks cover many activities, such as surveillance, epidemic intelligence, response, scientific consultation, research on microbiology, public health training, international relations, communication and awareness.

Since the first European outbreaks in 2020, the ECDC has been monitoring the progression of COVID-19, providing weekly reports about number of cases, case/test ratio, cumulative cases, incidence, hospitalizations and mortality in all of Europe and in specific countries. Each report specifies the changes that have been observed in the various countries in the last week regarding all the aforementioned parameters and summarizes the situation in each country. The data are accessible and can be downloaded into an excel file. Table 1 shows the ECDC variables currently available in the Atlas.

### 3.2. Organization for Economic Cooperation and Development

The Organization for Economic Cooperation and Development (OECD) is an economic organization founded in 1961 to stimulate economic progress and world trade [10]; it currently has 37 member countries. The member countries have generally high-income economies with a high Human Development Index, and through OECD, they provide a platform to compare policy experiences and coordinate domestic and international economic policies. The OECD performs various activities, such as creation and publication of tax conventions, the creation of the OECD Guidelines for Multinational Enterprises (i.e., a set of recommendations and standards for business conduct for multinational corporations), publishment of books, reports, statistics, papers and organization of conferences and events. Among these activities, it also provides an online portal where economic data can be freely consulted. Data are gathered for each member country with annual, quarterly, or monthly temporal resolution, allowing variations in economic indicators, such as GDP, import-export, employment and salaries in time.

The PERISCOPE Atlas includes several indicators coming from this dataset, so to visualize and monitor the possible economic changes due to the progression of the pandemic (See Table 1). The data are collected with different temporal granularities, usually monthly or quarterly.

For each variable of the Economic OECD data, several indicators can be visualized, i.e., absolute numbers, growth rates in relation in the same period (i.e., the last trimester/month of OECD data) and the previous period (i.e., the difference between the last trimester/month and the previous one). In the Atlas map, variables can be usually visualized with different modalities, such as polygons or points.

### 3.3. Policy Intensity Scores

Most of the sociopolitical data collected in PERISCOPE have been gathered thanks to a collaboration with the CoronaNet Research project [11,12]. This project started in late March 2020 with the principal aim of collecting information about the actions and interventions taken by national governments and subnational policy makers in response to the pandemic. CoronaNet’s dataset is the largest in the world to contain this kind of information, as it contains detailed data about policies’ typology, initiator, place, target, timing and enforcement methods. The data also include subnational data for some countries, which can be used to access and analyze public health policies that have been created by local governments and policy makers.

While the detail provided by CoronaNet’s event data can be useful for analyzing the drivers and effects of specific policies (e.g., closure of restaurants and retail stores in particular), it may be difficult to use for conducting research on the drivers and effects of general policy areas (e.g., restrictions of businesses more widely). To overcome these difficulties, Kubinec et al. [13] developed a new set of indices, Policy Intensity Scores, which measures the level of policy investment a government makes in a given policy domain that maximizes policymaker utility constraints. Under this framework, policy makers make tradeoffs between relative costs and benefits between COVID-19 suppression policies and other non-policy goals, which can include the financial, political or administrative ramifications of these policies. This model-based index is estimated using data from both the CoronaNet data and the Oxford COVID-19 Response Tracker [14], to produce indices which range from 0 to 100; high scores represent high levels of policy investment in a given policy area. These indices have been created for six policy areas, specifically:General Social Distancing, i.e., policies that seek to restrict mobility, including, e.g., lockdowns, curfews and travel bans.School Restrictions, i.e., policies that regulate the provision of education. They can include, e.g., school closures, remote teaching, hygienic policies in classrooms, etc.Business Restrictions, i.e., policies that regulate private commerce and industry, including closure of businesses, restrictions of customers allowed in stores, etc.Health Monitoring, i.e., policies which monitor individual health status efforts put into actively monitoring the disease through tests and other resourcesHealth Resources, i.e., policies which capture medical resources employed to treat the disease. These can include increased hospital and extra-hospital resources to face the burden caused by the virus.Mask Policies, i.e., policies that seek to restrict the spread of the virus by mandating or recommending facial coverings for certain people or places.

The indices are calculated for all EU countries (and are generally available for 180 countries more widely). They are available in the Atlas for visualization and analyses.

### 3.4. Other Data Sources

Besides the large amount of data already processed, other data sources have been identified for future integration in the Atlas. These include global open source data, local data sources and partnerships with other projects. The main supplementary data sources are:**Stockholm Karolinska Institutet**: this dataset is a pilot to test the feasibility of including aggregated mental health care utilization in the Atlas [15]. It contains weekly counts of mental health care utilization (primary and specialized care separately), stratified by age, for all individuals in Stockholm county during the first 6 months of 2019 and 2020, respectively. Mental health care utilization data is divided into whether consultations were physical (‘face-to-face’) or at a distance (e.g., by. phone, email or video calls), and whether they were new visits (defined as first psychiatric care contact within 365 days) or re-visits.**The 4CE Project**: this project is a vast international commitment d to perform electronic health record (EHR) data-driven studies of the COVID-19 pandemic, with the goal of informing doctors, epidemiologists and the general public about COVID-related data from local healthcare systems [16]. The consortium of this project is formed by a global community of researchers, most of which are members of the i2b2 Academic Users Group, which manages, computes and shares data extracted from EHRs of 96 hospitals across five countries: the US (45), France (42), Italy (5), Germany (3) and Singapore (1). The data includes daily counts by country, demographics of the hospitalized patients, case rates and reports about laboratory tests representative of renal function (creatinine), systemic inflammation (C-reactive protein), coagulopathy (D-dimer), liver function (total bilirubin) and immune response (white blood cells count). Lab data can be visualized by country or by site. Some of the partners that are involved in the 4CE project are also partners of the PERISCOPE consortium, namely Assistance Publique-Hôpitaux de Paris and the University of Pavia, which works in collaboration with the IRCCS Maugeri institution.

The integration of 4CE data will allow to visualize and predict clinical patterns, identify patient subgroups and improve patients monitoring. In particular, the events that most typically happen before, during and after a COVID-19 hospitalization can be identified and characterized, starting from general patients’ characteristics, such as previous and/or gained chronic conditions (modeled using the international ICD codes), and counting how many patients follow a specific pattern and which are their peculiarities. Figure 1 provides a schematic of this process.

**Google Mobility**: since the start of the pandemic, many international companies have invested a lot of resources to contribute to monitoring and contrasting the effects of the virus. Among them, Google has provided free access to mobility data [17], showing changes in populations’ movements, considering those users that allowed tracking by GPS. These data can be useful both to observe how the pandemic impacted mobility patterns and life habits, as well as to find possible trajectories of groups of people that can be used to model the spread of the disease. The data are usually available at a country level, subnational subdivisions are available for some countries (e.g., Italy and the United States). The raw data contain, on a daily basis, the percentage variation of movements towards specific categories of places, namely retail and recreation, grocery and pharmacy, parks, transit stations, workplaces and residential places. The baseline which the movements are compared to is the median value for the corresponding day of the week, during the 5-week period between 3 January and 6 February 2020, when there were no restrictions in any country except China.**Facebook Data**: besides Google, Facebook also initiated a data collection and visualization program to track the progression of the pandemic. This project, named *Data for Good* [18], gathers information about population density and movements to help researchers and public health authorities to assess the progression of the disease, plan interventions, vaccinations, policies, etc., and observe whether or not the specific interventions and policies are having an effect on flattening the curve of the disease. Besides COVID, other data are gathered to study phenomena such as climate change and business trends. Specifically, the data of interest for PERISCOPE is part of the so-called Coronavirus Disease Prevention Maps, which use anonymized and aggregated geo-localized data collected by specific surveys to calculate country-wise statistics on topics such as vaccination acceptance over time and the trend of personal preventive behaviors, such as mask wearing, social distancing, handwashing, etc.**Local Data Sources**: whenever available and useful, future datasets to be integrated in PERISCOPE also come from local sources, i.e., data collected at a regional or subregional level. One example of these data sources is the *Italian National Institute of Statistics (ISTAT)* [19], which collects subregional data about several indicators in Italy, such as the demographic situation, sociopolitical indicators, environmental changes, healthcare, criminality, economic situation, etc. Data of this kind can be used to create additional monitoring measures of the sociopolitical and economic situation in some areas, introducing parameters such as births/deaths trends, crime rate, changes in the consumption of some goods, etc.

## 4. The COVID-19 Data Atlas

This section illustrates the features of the first release of the Atlas, created in October 2021, currently available online at https://atlas.periscopeproject.eu. Accessed on 19 July 2022.

The Atlas as seen in the web User Interface is the result of a complex architecture that starts with the collection and integration of heterogeneous data that has been stored in a dedicated repository through specific backend services. Other services are then used to create visual analytics, advance analytics and the WebGIS platform, which, as explained in the rest of this section, are the main constitutional elements of the Atlas. Figure 2 shows a schematic representation of the Atlas architecture.

From the point of view of software components, the system is based on a relational PostgreSQL database, organized as a data warehouse. Visualization and querying are allowed directly from the WebGIS and from other user interfaces through a set of REST APIs. The whole architecture is deployed in the Amazon Web Services (AWS) cloud, to ensure scalability in terms of size and performance, service continuity and disaster recovery, data security, availability and long-term sustainability of the Atlas, even after the end of the project’s third year. In particular, the Atlas is fed by periodical ETL processes developed using cloud-native technologies (i.e., Amazon Glue), pulling preprocessed data, after appropriate data cleaning and curation, from a specially built, scalable data lake built on Amazon S3 service. The data lake is accessible, through appropriate access control, by all the partners of PERISCOPE, enabling them to develop procedures for data curation and loading on the Atlas data lake. Figure 3 shows a representation of the software components’ architecture.

The web User Interface of the Atlas starts with a map of the world, and features several simple commands. The Atlas can be subdivided into three main components: the WebGIS map, the Multivariate and Temporal Analytics sections and the Ontology. Figure 4 shows the initial Atlas interface as presented to the users.

### 4.1. WebGIS Map

The WebGIS Map allows to visualize and explore the data collected in the Atlas database in a classic GIS way, i.e., with layers that can be activated, deactivated and overlapped on a geographic map. In particular, a menu on the right side of the page allows to add layers of data, navigating them in a directory-like system where they are categorized according to data source and data type. Once a layer is selected, the relative data, aggregated by country, is visualized on the map. The visualized data can be navigated in space, selecting a specific country, and in time, setting a time filter that allows to select a specific period of time in a range that depends on the selected data.

In order to facilitate the comparison between variables, a second map can be enabled, allowing a side-by-side visualization of two different layers, that can provide visual information about the possible connections between different phenomena. Figure 5 shows a screenshot of the Atlas WebGIS with the side-by-side visualization functionality enabled. The time reference of the data can be modified in both maps separately.

### 4.2. Multivariate and Temporal Analytics

Comparative interactive visual analytics of a higher number of explored metrics/variables (than maximum possible through the abovementioned side-by-side twin WebGIS maps) in parallel is supported by additional different types of increasingly complex advanced dashboarding UI controls and visualizations, down the hierarchical drill-down-type workflow co-designed with relevant stakeholders (researchers, policy makers, etc.) and presented on Figure 6.

In this first release of the Atlas, the basic initial interactive multivariate analytic visualization type is the radar/spider diagram shown in Figure 7, commonly used and intuitively understood in social sciences, mental health research and related disciplines for exploring up to 10 variables simultaneously.

Specific efforts in the co-design process have generally been dedicated to jointly tailor the design and UX of all the Atlas rich dashboarding visualizations as intuitive and familiar as possible to the targeted users, while maintaining clear and uncluttered presentation of multivariate spatio-temporal data. Modularity of the UI and re-use of components and controls has also been worked out extensively; therefore, the data visualized on the spider diagram are controlled via the stack of interactive filtering panels on the left side (temporal, spatial or explored variables and policies scopes) and the timeline slider with play/stop animation, all common and applied also on other different visualization types.

Together with the Multivariate Analytics page, a Temporal Analytics page is also available, showing other kinds of representations that better clarify the time variations of the data, allowing to apply a set of different regression/regularization methods and to filter the type of visualized data and the time period of interest. An example is provided in Figure 8.

### 4.3. Ontology and Semantic Integration

The PERISCOPE semantic integration is based on an ontology aimed to be the reference knowledge model and interoperability resource for data dictionary, data mapping and harmonization. The ontology has seven high-level classes: the policy decision class represents the government responses to the coronavirus, and is applied to a specific geographical target (e.g., a specific region of a country) and a demographic target (e.g., population with high risk, e.g., elderly people). The policies are monitored by the enforcer and they will regulate specific services (commercial activities, public services, education, etc.) and type of places (public and private places). The last high-level class is the impact metrics that quantify the pandemic situation over three domains: health (e.g., epidemiological statistics, usage of health resources), socio-economic data (e.g., GDP, consumer prices) and metrics in relation with policies (e.g., Policy intensity Score). The policies have a double relationship with the impact metrics: on the one hand, policies take action over the current pandemic situation that can be described with the health impact metric, and on the other hand, the impact of the policy can be measured, using the metrics that refer to another specific temporal context (e.g., after 15 days the policy measure took place). The ontology has been co-designed with a multidisciplinary team of experts in policy making, social science researchers, economists, medical team, data science researchers and experts in computer science. The ontology will be linked with the data and provide unique definitions inside the Atlas data warehouse. It will be also used to add a semantic layer of the API services (thanks to the specification of the context using the JSON-LD format). When the integration will be completed, the research team will be able to provide to the public community an open dataset semantically enriched with the reference of the PERISCOPE ontology. The ontology is currently available at: http://periscope.lst.tfo.upm.es/. Accessed on 19 July 2022.

### 4.4. Data Modeling Sandbox

As previously introduced, besides being a visualization tool of a collection of high-quality data, the Atlas is intended to be a valuable source of specialized data analysis and machine learning (ML) models applied to COVID-19 data on policies and pandemic’s impacts. Some of these analyses have been already initiated at the time of writing, and they are applied both to global scale data and to smaller geographical realities, usually determined by specific connections that the PERISCOPE consortium has with local data providers, for example hospitals and municipalities.

In order to provide highly specialized advanced analytics and modeling/ML capabilities to users involved in the analytics, a “modeling sandbox” has been designed, where users can access data stored in the Atlas, or even semi-structured data collected in the data lake, through a fully-hosted web interface enabling early access to data being collected by PERISCOPE. The data modeling sandbox relies on state-of-the-art technologies that are widely used in the data science and statistics fields offering a number of facilities, such as file management, remote batch execution and support to data science languages such as Python, R, Scala and Julia.

## 5. Discussion

The PERISCOPE Atlas, as has been presented in this paper, is in its first release, which was the result of the construction of a complex data integration architecture that has been exploited creating some basic functions and results. Thanks to the creation of this architecture, it will be easy to extend the Atlas with new data and new functionalities in the incoming months, creating a vast tool that can be a starting point or a reference for research projects and health policy making.

Since the pandemic has pushed global research, the PERISCOPE Atlas is not the only tool that will be used to monitor the progress of the disease or to analyze its impacts in view of future prevention strategies, many data repositories have been created, and research over topics such as the effects of the pandemic in the last years has progressed quickly. Public dashboards such as the ones cited previously in this paper have been created and made available to the general public, and international companies such as Facebook and Google have started an impressive data collection procedure to analyze several aspects of the pandemic, such as changes in populations’ movements and socioeconomic implications. This global effort to collect and share data is partially the direct result of a paradigm that has been established in the last decades, with the advent of big data analysis and of new technologies that make the processes of data collection, analysis and sharing much easier than before. Many European projects in the past have already focused on these tasks, creating accessible repositories of medical data and analysis tools such as the AmsterdamUMCdb database, which contains data concerning intensive care units across all of Europe [20]. Even the sources used in PERISCOPE, such as ECDC and OECD, created data access tools long before the pandemic started.

While on one hand, this fast-growing availability of data and technologies makes it easier for researchers and policy makers to access and analyze public health data, on the other hand, the creation of multiple data repositories makes it sometimes difficult to find the needed data, and it creates the necessity to perform complex data integration and preprocessing procedures, as the heterogeneity of data formats and characteristics can be challenging. This is especially true for COVID data, as the collection of data was so fast that repositories, dashboards and analysis tools were created in an unprecedented short time. For this reason, one of the innovation points of the PERISCOPE Atlas stands in the integration of several datasets concerning different aspects of the pandemic, which will allow researchers, policy makers and the general public to quickly access high volume of heterogeneous pre-processed data that would otherwise need to be found in fragmented sources. Furthermore, thanks to the link between the data modeling sandbox and the Atlas, results of complex statistical and Machine Learning models can be presented in a way that is intelligible even to users that are not expert in the field, overcoming another frequent limitation of current research, i.e., the lack of availability of information for those who do not have expertise in the research field.

One point of particular interest for the PERISCOPE project is the development of complex machine learning and statistical tools to explore possible connections between variables and create predictive models that can be both used to examine the progression of the pandemic and reused in case of new public health emergencies that might rise in the future. At the time of writing, several studies are already in place within the PERISCOPE context, and their results will be released in the Atlas as soon as possible. Particular focus is given to the Policy Intensity indexes, which are an innovative representation of the sociopolitical response of a country to the virus and, being a numerical score, can be easily used to compute correlations or develop complex machine learning models to compare countries or policies, find intervention patterns and predict possible policy outcomes.

Numerous innovative models and analyses have already been developed and tested in the context of the PERISCOPE project, and the sandbox will allow to make their results available to researchers, policy makers and the general public. Generally speaking, these models focus on studying and predicting the temporal evolution of the pandemic and its impact on the world socioeconomic panorama. For instance, Autoregressive Models of different kinds (e.g., Poisson regression, Bayesian models) can be used to model the case rate in relation with public health containment policies [21,22,23], creating a predicting tool to understand contagion trends and identify the best containment measures. Generally speaking, the results of these implementation have shown to be country-specific. Statistical methods, such as Principal Component Analysis (PCA) or Dynamic Factor Models (DFM), allow to create statistical models of epidemiological susceptibility risks around the world [24], providing a further instrument for policy makers and organizations to assess the epidemiological risk during epidemics and predict possible consequences. Statistical an ML analysis has been performed also on economic data, for example in Ahelegbey et al. [25], where network analysis has been used to study the connections among several companies before and during the pandemic, showing that there were important changes reflecting the economic difficulties caused by the spread of the virus.

## 6. Conclusions and Future Developments

With the outbreak of the COVID-19 pandemic, researchers from all across the world began an effort to face the consequences of the diffusion of the virus. In the last two years, research institutes and private companies have directed resources to study the transmission and progression of the disease, and also to model the socioeconomic implications [26,27,28]. Among all these, the PERISCOPE project brought together a heterogeneous collection of research partners with various expertise in clinical, technological, economic and sociopolitical environments. This has allowed for the creation of comprehensive tools that merge data and information coming from different contexts, allowing to perform complex multi-level analysis on all the impacts of the pandemic.

One iconic example is the PERISCOPE Atlas, which serves mainly as a visual representation of the integrated data and the results of statistical and machine learning analyses that can be performed on them, with the aim of finding important association among different variables, such as health monitoring variables and public health policies, which can be used to improve the management of the pandemic and increase the preparedness to face new ones.

This paper presents the first release of the Atlas, characterized by a complex backend architecture that allows for high-quality heterogeneous data ETL and integration, and different specific tools for data visualization and analysis, such as the WebGIS and the temporal/multivariate analytics. Through additional advanced systems such as the data modeling sandbox, future releases will feature also more complex statistical and machine learning models that can be used to better inspect all the impacts of the pandemic in a detailed way. More data integration will be performed as well thanks to the extension of the ETL system that has been already created.

As the PERISCOPE project continues, the Atlas will be extended adding new data and new functionalities. In particular, more data coming from local sources, such as those listed in Section 3, will be integrated through the backend services already in place, allowing to access detailed information with a subnational spatial granularity for some European cities or regions. More temporal and multivariate analytics will be added as well, taking into account the addition of new data and the progression of the pandemic, which constantly introduces new variables, such as vaccinations and new variants, that can change the causality mechanisms underlying the relations among health, policy making and economics. Other kinds of data, for example behavioral data, will be added as well.

## Figures and Tables

**Figure 1 ijerph-19-09136-f001:**
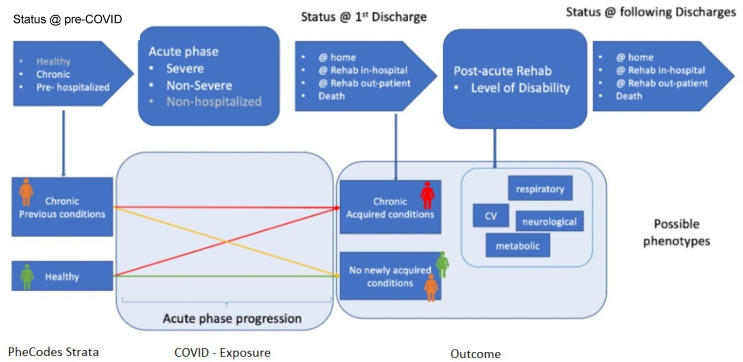
Scheme of the 4CE process that can be used to predict clinical patterns and identify subgroups of patients.

**Figure 2 ijerph-19-09136-f002:**
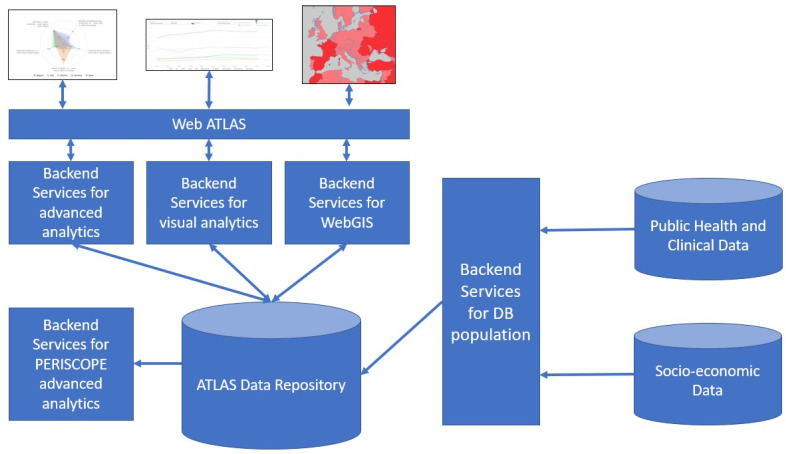
Data collection, integration and elaboration process in the PERISCOPE Atlas.

**Figure 3 ijerph-19-09136-f003:**
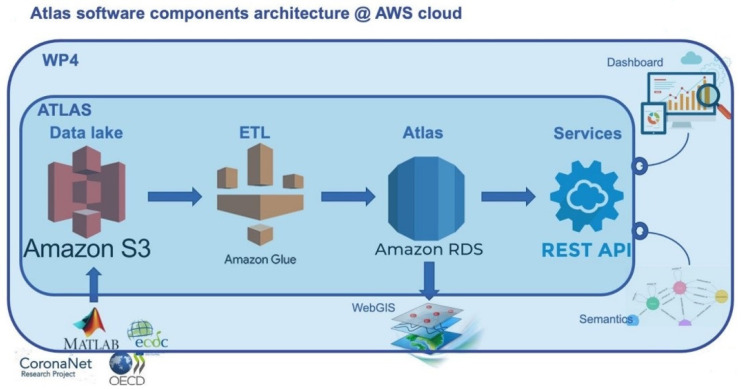
Software components of the Atlas.

**Figure 4 ijerph-19-09136-f004:**
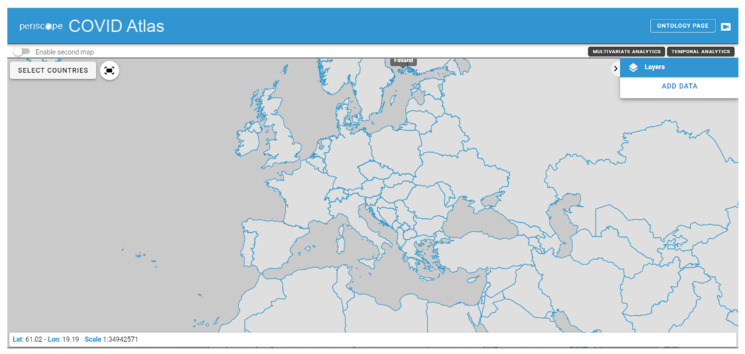
Initial Atlas interface.

**Figure 5 ijerph-19-09136-f005:**
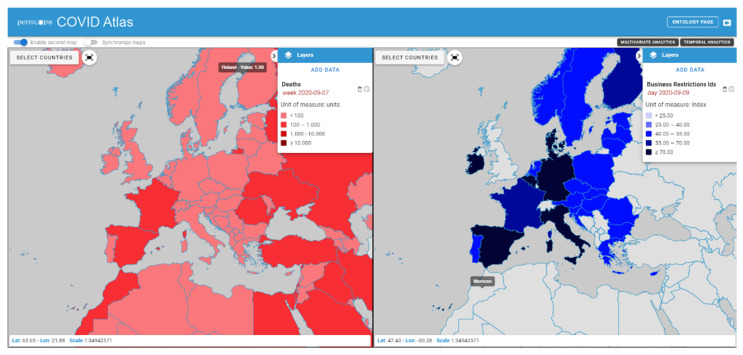
A sample page of the Atlas WebGIS with side-by-side visualization of two maps, one showing the number of deaths (**left side**) and the other showing the Business Rrestiction Index (**right side**) for the same week.

**Figure 6 ijerph-19-09136-f006:**
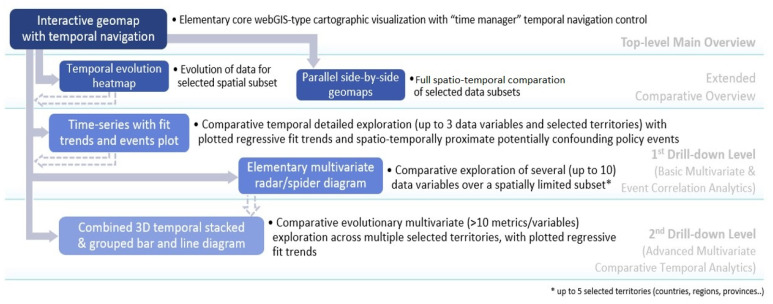
Atlas analytics workflow.

**Figure 7 ijerph-19-09136-f007:**
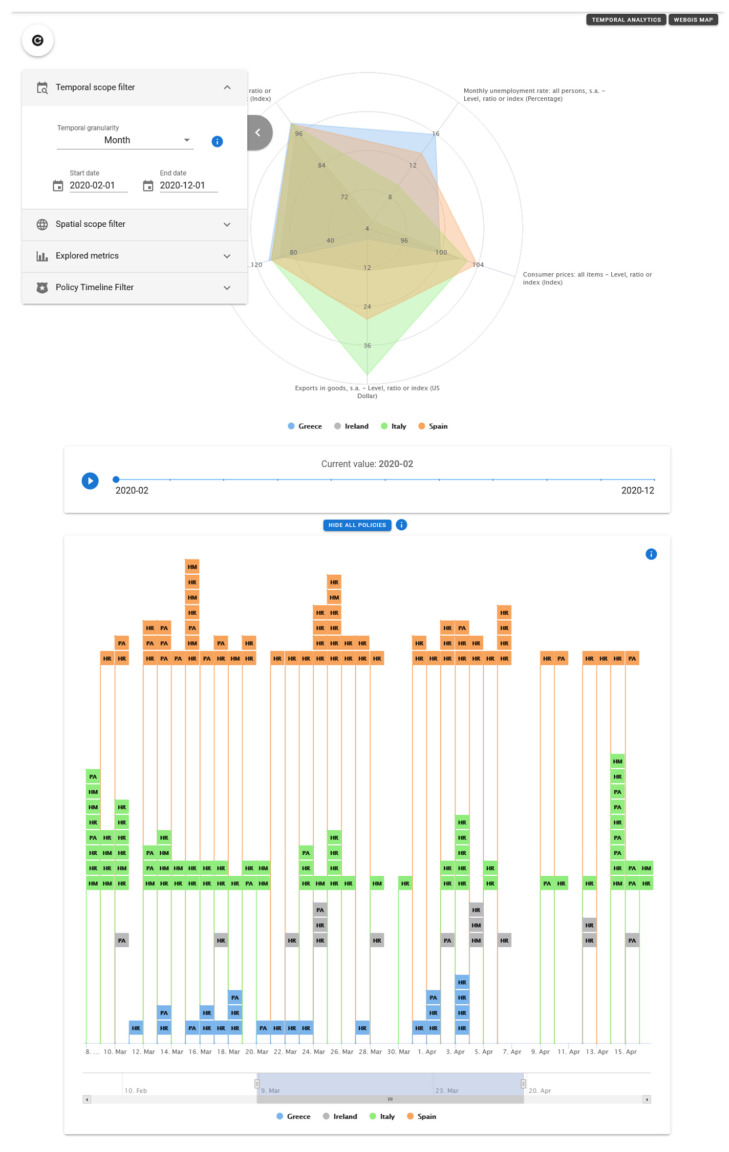
Example of visualization of the Multivariate Analytics page of the Atlas.

**Figure 8 ijerph-19-09136-f008:**
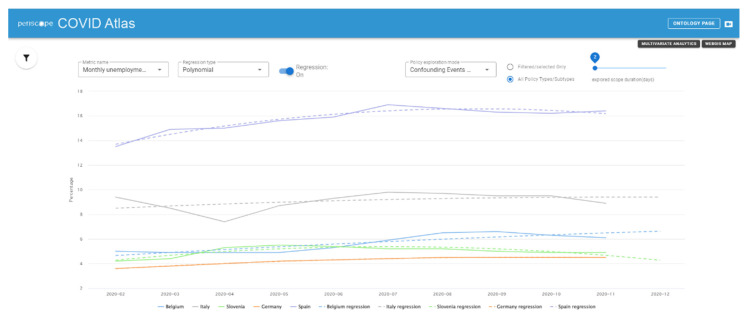
Example of the visualization in the Temporal Analytics page of the Atlas.

**Table 1 ijerph-19-09136-t001:** Data currently available in the Atlas for visualization and analysis.

Data Type	Data Source	Variables
Health Data		Weekly Cases Weekly Case Rate
	ECDC	Weekly Hospital Occupancy Weekly Hospital Admission
		Weekly ICU Admission Weekly Deaths
Economic Data		Broad Money
	OECD	Consumer Prices Export in Goods GDP Hourly Earnings Imports in Goods Industrial Production Passenger Cars Registrations Permits Issued for Dwellings Retail Trade Share Prices Total Employment Unit Labor Cost
Socio-political Data	CoronaNet Policy intensity Scores	Business Restriction Index Health Monitoring Index Health Resources Index Mask Policies Index School Restrictions index Social Distancing Index

## Data Availability

Several publicly available datasets have been already integrated inside the Atlas and are cited in this paper. ECDC data about cases and deaths can be found at this link: https://www.ecdc.europa.eu/en/publications-data/download-historical-data-20-june-2022-weekly-number-new-reported-covid-19-cases, ECDC data on hospitalizations and ICU occupancy can be downloaded here: https://www.ecdc.europa.eu/en/publications-data/download-data-hospital-and-icu-admission-rates-and-current-occupancy-covid-19. OECD data can be browsed and downloaded here: https://data.oecd.org/. Policy Intensity Indexes can be found here: https://github.com/CoronaNetDataScience/corona_index.

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
