# Peer review of "A New Interactive Tool to Visualize and Analyze COVID-19 Data: The PERISCOPE Atlas"

_ijerph, 2022, doi:10.3390/ijerph19159136_

Round 1
Reviewer 1 Report
Many thanks to the authors for this interesting article. It is well written and gives other researchers a good overview about the PERISCOPE project and the contents and functions of Atlas in particular.
* As the article is essentially a description of the periscope project (and I believe the authors described it as it is), I have no concerns regarding the methodology or other aspects that would apply for other original research articles.
* Most readers should find the paper easy to understand, even if they do not have advanced computer or data science skills. However, the paper would be suitable for a wider audience if some aspects were described in more detail/with more background. Figures 2 and 3 in particular could be a little more "self-explanatory".
* At some points in the article (especially with the ML models) it would have been interesting to know a little more about what research is already running. However, I understand that this is not the main focus of the article, which is on the general description of PERISCOPE and Atlas. Nevertheless, the authors might consider providing a few more details about ongoing research projects and research questions to be answered in sections 5 and 6. So far, the descriptions in the paper are rather general.
* The figures seem to be screenshots. Especially those from PPT or Word (figures 1 and 6) need to be fixed because they still contain the autocorrect annotations (underlinings). Generally, the figures should be replaced by high resolution images, as the current image quality is not very good. However, this will most certainly be part of the further publication process.
Author Response
We thank the Reviewer for the compliments and the useful suggestions. It is true that the main focus of the paper is a general description of the Atlas, its functionalities and its potentialities, therefore we did not focus on describing running research and Machine Learning tools used. Nevertheless, we appreciate the Reviewer’s suggestion to write something more about these topics, in order to widen the audience for this paper.
For this reason, we added a new Discussion section, containing some references and a quick description of some work that has been and is being performed in the project. The previous future development part has been moved with the conclusions.
We also fixed the Figures like suggested and we thank the Reviewer for helping us notice that they needed to be improved.
Reviewer 2 Report
The article presents preliminary outcomes of the Periscope project. In particular, it describes the current status on a web based visualization and analysis component called Atlas. While it is good to inform the international community of ongoing efforts, the content in the article is not based on a clear scientific question, and it thereby also does not benchmark itself against relevant related work. Furthermore, the article leaves it too implicit which specific integrations are truly new, what is the EU coverage, where are the current limitations in terms of data type coverage per country, and what is the roadmap to overcome limitations. Implicitly, much of the EU coverage appears to be based on the progress in other EU projects, like CoronaNet and efforts by long-term efforts by the ECDC and the OECD.
I strongly encourage the authors to focus on sharing their preliminary results in general media, perhaps and international workshops, yet rethink which are the fundamental scientific questions being answered before considering again scientific journals as the publication outlet.
For a scientific journal publication, please discuss more explicitly the previous efforts in a given discipline, clarify where the state-of-the-art has significant limitations, argue why certain research methods make most sense, share the results and also point to specific limitations that still need to be addressed. Please also include a discussion reflecting on the choice of the research methods, such that other scholars can learn from it.
Then, there are also quite detailed inconsistencies one could identify in the article (e.g., in the abstract, where first "socioeconomic, political, behavioral and economic effects" are mentioned and then "health, economic and sociopolitical data", which makes one wonder why behavioral data would not be included), but such details are secondary to the more fundamentally lacking focus on a clear and novel research question.
When considering the related work, please do look beyond the COVID19 pandemic, and also discuss the progress that was made in decades of data integration projects across the EU and globally. Please leverage such literature to be clearer about what was so new about the COVID related data. When discussing COVID specific efforts, please do include also initiatives such as https://github.com/AmsterdamUMC/AmsterdamUMCdb, where much richer data is being integrated and shared.
Finally, please consider including a strong "running example" to demonstrate an analysis which was previously impossible. Such an example first makes matters more concrete, and second it can also inspire and enable others to leverage your infrastructure.
Author Response
We thank the Reviewer for the interesting comments. We realized that it is true that there was not a clear research question and that a proper context concerning the state of the art of data integration tools was not provided. Following the Reviewer’s suggestion, we added a paragraph in the introduction (lines 60-69) and we modified sections 5, that is now entitled “Discussion” (we moved the future developments with the conclusions). We added some citations of related work, like the one suggested by the Reviewer, we did not add many references to other work related to Covid data and dashboards, as the ATLAS itself is a collection of different sources that are described in the manuscript, and one of the innovative points stands in this. As we wrote in the new Discussion section, the combination of the incredible advancements of the last years in the field of data analysis and the necessity to deal with the pandemic lead to the creation of numerous different tools and projects that allow to access and analyze data related to all aspects of the pandemic. The strength of the PERISCOPE ATLAS is not in the collection of data itself, but it stands in mainly two aspects:
- It is a heterogeneous collection of different sources, where data and analyses about several aspects of the pandemic impacts can be found without the need of searching separate sources. In fact, one of the drawbacks of the fast data collection and analysis of these years is fragmentation, the collected data is a lot, scattered through various repositories and tools, and the collection of heterogeneous information regarding a specific topic can be a long process. The ATLAS allows to easily perform this operation. Plus, different levels of analysis can be performed, from simple data visualization to advanced machine learning, with results that are reported in a way that can be understandable even for a non-expert audience.
- The data modeling sandbox provides the possibility to perform different innovative machine learning ad statistical analyses and integrate their results with a powerful visualization tool that is also part of the ATLAS. Examples of these analyses have been provided at the end of the Discussion section. To our knowledge, this is a unique feature among all the Covid data tools available and provides researchers and policy makers with a powerful instrument that allows to collect fast and detailed information and ideas.
We also thank the Reviewer for pointing out that we did not cite behavioral data in the manuscript, we added a sentence about it in the future developments, as there is currently no behavioral data in the ATLAS but we are planning to add some in the near future (for example from Google or Facebook).
Concerning the EU coverage and data type limitations, the ATLAS includes data from several countries, even extra-EU ones, and limitations vary depending on the specific country and data type, as the data collected and the space and temporal granularities vary a lot from one country to another, sometimes they vary even within different subregions. Therefore, it would be difficult to provide detailed data about all the limitations concerning each single country.
Reviewer 3 Report
This paper introduces PERISCOPE Atlas: an interactive web app to visualize and analyze COVID-19 data. Minor details to point out, mainly about the Figures shown ( please check attached document ). Overall it is an interesting proposal and could be very useful if properly used.

Author Response
We thank the Reviewer for the appreciation of our work and for giving useful comments and suggestions. We made the proposed modifications and hope that the revised version looks better now.